# Distinct Expression Patterns of VEGFR 1-3 in Gastroenteropancreatic Neuroendocrine Neoplasms: Supporting Clinical Relevance, but not a Prognostic Factor

**DOI:** 10.3390/jcm9103368

**Published:** 2020-10-21

**Authors:** Florian Bösch, Annelore Altendorf-Hofmann, Sven Jacob, Christoph J. Auernhammer, Christine Spitzweg, Stefan Boeck, Gabriele Schubert-Fritschle, Jens Werner, Thomas Kirchner, Martin K. Angele, Thomas Knösel

**Affiliations:** 1Department of General, Visceral, and Transplant Surgery, Ludwig-Maximilians-University Munich, 81377 Munich, Germany; florian.boesch@med.uni-muenchen.de (F.B.); Sven.jacob@med.uni-muenchen.de (S.J.); jens.werner@med.uni-muenchen.de (J.W.); martin.angele@med.uni-muenchen.de (M.K.A.); 2Interdisciplinary Center of Neuroendocrine Tumors of the GastroEnteroPancreatic System, Ludwig-Maximilians-University of Munich, 81377 Munich, Germany; Christoph.Auernhammer@med.uni-muenchen.de (C.J.A.); christine.spitzweg@med.uni-muenchen.de (C.S.); stefan.boeck@med.uni-muenchen.de (S.B.); gabriele.schubert-fritschle@med.uni-muenchen.de (G.S.-F.); Thomas.Kirchner@med.uni-muenchen.de (T.K.); 3Department of General, Visceral und Vascular Surgery, Friedrich-Schiller University, 07743 Jena, Germany; annelore.altendorf-hofmann@gmx.de; 4Department of Internal Medicine 4, Ludwig-Maximilians-University Munich, 81377 Munich, Germany; 5Department of Medicine 3 and Comprehensive Cancer Center, Ludwig-Maximilians-University Munich, 81377 Munich, Germany; 6Munich Cancer Registry (MCR) of the Munich Tumour Centre (TZM), Institute for Medical Information Processing, Biometry, and Epidemiology (IBE), Ludwig-Maximilians-University Munich, 81377 Munich, Germany; 7Institute of Pathology, Ludwig-Maximilians-University, 81337 Munich, Germany

**Keywords:** neuroendocrine neoplasia, VEGFR, PD-L1, PD-1, microarray

## Abstract

Introduction: Gastroenteropancreatic neuroendocrine neoplasms (GEP-NENs) are an increasing tumor entity. Since many patients are diagnosed at an advanced stage, treatment is still challenging and dependent on many tumor and patient specific factors. Therefore, the aim of the present study was to elucidate the expression rates and the prognostic value of vascular endothelial growth factor receptor (VEGFR) 1-3 in GEP-NENs. A potential association to immune checkpoint markers was further investigated. Material and Methods: The expression levels of VEGFR 1-3 were analyzed by immunohistochemistry and correlated with the expression of the checkpoint markers PD-1 and PD-L1. Furthermore, the tumor samples of 249 GEP-NEN patients were studied and correlated with overall survival rates and clinicopathological features. Kaplan–Meier analyses and the log rank test were used for survival analyses. Categorical variables were compared by the χ2 test. Results: The most common primary tumor site was the small intestine (50.6%), followed by the pancreas (25.7%). VEGFR 1 was highly expressed in 59%, VEGFR 2 in 6.4%, and VEGFR 3 in 61.8% of the analyzed samples. The expression of VEGFR 1-3 was not significantly associated with survival rates. Pancreatic NENs had the highest expression of VEGFR 1 and 3 in 80% of the cases. VEGFR 1-3 positivity correlated with the expression levels of immune checkpoint markers. Discussion: VEGFR 1-3 show a distinct expression pattern in different subgroups of neuroendocrine neoplasias indicating a conceivable target. Moreover, there was a substantial association between VEGFRs and immune checkpoint markers. Taken together, anti-VEGFR therapy represents a promising therapeutic approach in GEP-NEN patients and should be addressed in future studies.

## 1. Introduction

Gastroenteropancreatic neuroendocrine neoplasms (GEP-NENs) gain increased clinical significance since their incidence is continuously increasing [1]. GEP-NENs are nowadays as common as testicular tumors, gliomas, and multiple myeloma [2]. Surgery remains the gold standard for cure [3,4], but GEP-NENs are often diagnosed at an advanced stage making therapy challenging [2]. Tumor grading directly influences the therapeutic approach and even stage IV patients with well to moderately differentiated tumors might have beneficial survival rates [5]. Biotherapy with somatostatin analogs is the first line of systemic treatment for well-differentiated GEP-NENs [6,7,8], for poorly differentiated neuroendocrine carcinomas (NECs) the use of systemic chemotherapy is recommended [9]. Targeted therapy has been evolved for the treatment of GEP-NENs [9,10]. Nonetheless, more research is needed in order to easily decide which therapy is best and to monitor treatment efficacy. Thus, optimal management of GEP-NEN patients is still challenging and predicting the therapeutic value of a single agent would be desirable.

Pathologic de-novo angiogenesis is a key step in cancer development since it facilitates local tumor progression and further leads to distant metastases [11,12]. Vascular endothelial growth factor (VEGF) is a key player in angiogenesis and fosters angiogenesis by binding to a distinct cell surface receptor (VEGFR 1-3). GEP-NENs are highly vascularized tumors and it was demonstrated that VEGF is expressed on GEP-NENs [13]. VEGF and VEGFR have been extensively studied and efforts have been made to inhibit the VEGF/R pathway. Therefore, the multi-kinase inhibitor sunitinib, targeting, inter alia, VEGFR, is approved for the treatment of advanced pancreatic neuroendocrine neoplasms (pNENs) [14]. However, treatment with sunitinib might lead to essential side effects and a more tailored therapeutic approach inhibiting for instance only one or two kinases might be less noxious [15]. In this respect, increasing evidence suggests that the combination of VEGF/R inhibitors and novel checkpoint inhibitors might be prognostic beneficial [16,17]. Nonetheless, a comprehensive analysis of the expression rates of VEGFR 1-3 and immune checkpoint markers has not been investigated so far. Moreover, the predictive value of the expression levels of VEGF and VEGFR is inconsistent [18,19] and a detailed analysis of the association of VEGFR 1-3 and its prognostic role in GEP-NENs has not been conducted so far.

Thus, the present study aimed to evaluate the expression rates of VEGFR 1-3 and to analyze a potential association to immune checkpoint markers in GEP-NENs. Furthermore, the prognostic value of these markers was investigated.

## 2. Material and Methods

### 2.1. Patient Cohort

In the present retrospective cohort study 249 formalin fixed paraffin-embedded GEP-NEN specimens from the Institute of Pathology of the Ludwig-Maximilians-University Munich were analyzed. Patients diagnosed with a GEP-NEN between the years 1995–2013 were included and clinicopathological parameters were retrieved from the pathologists’ original reports and a prospectively led database. In the present study, the median time between initial diagnosis and the specimen was obtained was four days. Primary tumor grading was established by the Ki67 index or the mitotic count. The tumors were reclassified regarding the current 2019 WHO classification [20,21]. Data on follow-up were attained from outpatient clinics and the Munich Cancer Registry.

The study was carried out according to the recommendations of the local ethics committee of the Medical Faculty of the Ludwig-Maximilians-University Munich, Germany, which approved the study protocol (project identification code: 18-177). Irreversibly anonymized data sets and tissue specimens were used for this retrospective analysis. The study was performed according to the standards laid out in the declaration of Helsinki 1975. All researchers were blinded to the patient data during the experimental analysis. 

### 2.2. Tissue Microarrays and Immunohistochemistry

The tissue microarrays (TMA) were assembled as described previously [22]. In brief, of each tumor specimen two punch biopsies with a diameter of 0.6 mm were stained and analyzed.

For the immunohistochemical staining commercially available antibodies against VEGFR 1-3 were used. VEGFR 1: Dilution 1:50, Ab32152, Abcam, Cambridge, CAM, United Kingdom; VEGFR 2: Dilution 1:100, 2479, Cell Signaling Technology, Danvers, MA, USA; VEGFR 3: Dilution 1:20, NCL-LVEGFR3, Novocastra, Wetzlar, HE, Germany. The expression levels of the immune markers programmed death-1 (PD-1) and of its ligand, programmed death ligand-1 (PD-L1) were tested with monoclonal antibodies (PD-1: Dilution 1:80; clone NAT105, Cell Mark, MEDAC, Wedel, SH, Germany; PD-L1: Dilution 1:100; E1L3N, Cell Signaling Technology, Danvers, MA, USA). Immunostaining was performed according to standard procedures as previously described [23,24]. Every TMA was counterstained with hematoxylin (Vector) and system controls were included to avoid unspecific staining.

Two independent and blinded observers (F.B. and T.K.) evaluated the immunohistochemical staining semi-quantitatively (0, negative; 1, weakly positive; 2, moderately positive; 3, strongly positive) (Figure 1 and Appendix A). As described previously, PD-1-expressing tumor-infiltrating lymphocytes were counted manually and categorized semi-quantitatively per punch into low (<3 positive lymphocytes) and into high (≥3 positive lymphocytes). High/positive expression of PD-L1 was defined as membranous staining of >1% of the tumor cells [23]. If there was discrepancy the slides were analyzed again under a multiheaded microscope and consensus reached. Due to the number of patients included, binary evaluation was decided. Therefore, GEP-NENs were either graded as no/lowly (score 0 and 1) or highly (score 2 and 3) expressing tumors.

### 2.3. Statistical Analysis

Statistical analysis was performed with the use of SPSS 22.0 (Armonk, NY, USA: IBM Corp.) software. Categorical variables were compared by χ2 test. Survival curves were calculated by the Kaplan–Meier method. The log rank test was used to assess differences in survival. Cox proportional hazard analysis was conducted to identify independent prognostic factors. *p*-values lower than 0.05 were considered statistically significant and *p*-values lower than 0.001 were considered statistically highly significant.

## 3. Results

### 3.1. General Clinicopathological Findings

A total of 249 patients have been analyzed, including 112 women (45%) and 137 men (55%). Median age was 61 years (18–92 years). The majority of 50.6% (*n* = 126) had a small intestine NEN (si-NEN), followed by 25.7% (*n* = 64) of patients with a pNEN. Grading was available for every patient included. The present study cohort includes 152 patients (61%) with a G1 GEP-NEN, 81 patients (32.5%) with a G2 tumor, and 16 patients had a G3 (6.5%) tumor. Due to the distribution with predominant G1 NETs the median in the tumor collective was 1% (1–90%). Reclassification revealed that 16 G3 GEP-NENs were included in the study collective. Six patients had a G3 NET and ten patients had a G3 NEC. Surgery represents the most common therapeutic approach (*n* = 224/244; 91.8%). Comparable to the broad diversification of grading, the primary tumors were T1 in 23.9% (*n* = 53/222), T2 in 20.7% (*n* = 46/222), T3 in 37.3% (*n* = 93/222) and T4 in 12.0% (*n* = 30/222) of cases. Distant metastases were seen in 103 patients (41.4%), most commonly to the liver (*n* = 72; 28.9%). Eleven patients (4.4%) had bone metastases. Lymph nodes were harvested in 186 patients and in 71% of these (132 cases) positive lymph nodes were documented. The main histopathological criteria are summarized in Table 1.

### 3.2. Correlations

VEGFR 1 was highly expressed in 59% (*n* = 147) and VEGFR 3 in 61.8% (*n* = 154). In contrast, VEGFR 2 showed no/low expression in the majority of patients (*n* = 233; 93.6%). There was a statistically significant correlation between the expression levels of VEGFR 3 and the expression levels of VEGFR 1 and VEGFR 2. There was no significant association of the expression levels of VEGFR 1 and VEGFR 2 (Table 2). A statistically significant correlation between the expression of VEGFR 1-3 and grading was not evident (VEGFR 1: *p* = 0.153; VEGFR 2: *p* = 0.274; VEGFR 1: *p* = 0.301).

Expression of VEGFR 1-3 showed a statistical significant correlation with tumor stage. High expression of VEGFR 1 and 3 was seen in 75.9% (*n* = 41) and 72.2% (*n* = 39) of cases without lymph node metastases, respectively. In contrast, 72.2% (*n* = 39) and 56.1% (*n* = 74) of cases with a high expression of VEGFR 1 (*p* = 0.019) and 3 (*p* = 0.048), respectively, had lymph node metastases. In patients with distant metastases high expression of VEGFR 2 was seen in 12.5% (*n* = 2) in contrast to 87.5% (*n* = 14) of patients without distant metastases (*p* = 0.017).

Patients were categorized according to the site of their primary tumor into pancreatic and non-pancreatic GEP-NEN primary. The expression levels of VEGFR 1 and VEGFR 3 showed a statistically significant association with the primary tumor location. In this respect, 80% (*n* = 52) of pNENs were VEGFR 1 and VEGFR 3 positive, respectively (*p* < 0.001). VEGFR 2 expression levels were not associated with a specific primary tumor.

A further analysis revealed that VEGFR 2 correlated significantly with the expression levels of the immune marker PD-1 (*p* = 0.004) (Table 3).

Its ligand, PD-L1, had a significant association to the expression of VEGFR 3 (*p* = 0.001). VEGFR 1 expression levels showed no significant correlation to immune checkpoint markers (*p* = 0.643) (Table 4).

### 3.3. Survival Analyses

Five- and 10-year survival rates of the entire cohort were 74% and 58%, respectively. In univariate analysis observed long time survival was statistically significant influenced by age (*p* = 0.002), grading (*p* < 0.001), the resection of the primary tumor (*p* = 0.001) and the presence of distant metastases (*p* = 0.004). However, the expression levels of VEGFR 1-3 did not show a significant correlation with survival rates (Figure 2). Previously reported prognostic factors, such as grading, the presence of distant metastases as well as the site of distant metastases were included in multivariate analysis [25,26]. This analysis revealed that distant metastases were an independent risk factor. Grading and the presence of bone metastases represented further independent prognostic factors (Table 5).

## 4. Discussion

GEP-NENs are highly vascularized tumors, thus pathologic de-novo angiogenesis is an important characteristic feature of this entity. Furthermore, neuroendocrine cells of the digestive tract are capable of producing VEGF, which is mandatory for microvessel stability in GEP-NENs [13,27]. The coexpression of VEGF and its receptors in tumors seem to directly lead to a higher developed vascular architecture [28]. In this respect, the present study provides a detailed analysis of the most comprehensive patient collective regarding the expression of VEGFR 1-3 in GEP-NENs. VEGFRs are a family of tyrosine kinase receptors consisting of three members: VEGFR 1 or Flt-1 (fms-like TK1); VEGFR 2 or KDR (kinase insert domain-containing receptor TK); VEGFR 3 or Flt-4 (fms-like TK4) [29].

The expression levels of the three VEGFRs do not represent a prognostic factor. Nevertheless, the definition of its expression levels might be beneficial for certain patients since there was a significant correlation of the expression levels of VEGFR 1-3 with tumor stage. Treatment with multikinase inhibitors consists commonly of a combination with different regimens such as the inhibitor (bevacizumab, sunitinib) plus somatostatin analogs and chemotherapy, respectively [14,30]. This therapeutic approach is commonly associated with clinically relevant side effects and thus the adherence to the therapy might be reduced up to 54% [31]. Additionally, monotherapy with VEGFR-inhibitors also can lead to a wide spectrum of severe adverse reactions. Especially cardiovascular side effects are commonly seen and should be thoroughly monitored [32,33]. Decreased adherence in turn results in a reduced therapeutic effect. Thus, the results of the present study might help to define patients who might benefit from anti-VEGFR therapy. Subsequently adherence might increase. Moreover, by assessing parameters in CT scans a prognostic relevant distinction of patients seems to be feasible. This non-invasive approach is at least in part “dynamic” and allows evaluating the tumor heterogeneity over time without the need of a further biopsy [34]. Evaluating treatment efficacy seems possible and might further help to increase adherence.

There was no significant association of the expression levels of VEGFR 1-3 and overall survival. Nonetheless, these results are in line with previous studies on other gastrointestinal malignancies [35,36]. This might in part be explained by the fact that well- to moderately differentiated GEP-NENs have a favorable overall survival even in the presence of distant metastases and that there was no significant correlation between tumor grading and the expression of VEGFRs. Furthermore, neuroendocrine carcinomas seem to have a different VEGF homeostasis [15].

In our study VEGFR 1 and 3 were highly expressed in 59% and 61.8%, respectively, whereas only 6.4% of GEP-NENs showed VEGFR 2 positivity. Nonetheless, VEGFR 2 expression rates in GEP-NENs have been reported to be up to 60%. However, this study analyzed only 20 samples regarding the expression rate of VEGFR 2 [37]. Studies on small intestine NENs and other tumor entities, such as leiomyoma and soft tissue sarcoma, revealed also a difference in the expression rates of the three VEGFR receptors [24,38,39]. Moreover, since the majority of patients underwent surgery, a possible selection bias further influences the results additionally neglecting the potential influence of systemic treatment. Additionally, pNENs had a statistically significant higher expression of VEGFR 1 and 3 compared to other gastrointestinal NENs.

The formation of distant metastases of pNENs seems to be directly influenced by VEGF [40]. Thus, a therapeutic approach inhibiting this pathway appears to be promising. Therefore, the multikinase inhibitor sunitinib and the monoclonal antibody bevacizumab have both been evaluated in clinical trials including patients with advanced pNENs [41,42]. Sunitinib, which irreversibly blocks tyrosine kinases such as VEGFR 2 and 3 and platelet-derived growth factor receptor (PDGFR), showed a prolonged progression free survival compared to placebo [14]. Bevacizumab blocks binding to VEGFRs by specifically binding VEGF in the blood stream. Clinical trials demonstrated beneficial effects of bevacizumab in the treatment of pNENs [30,43,44]. Taken together, the results of the present study may at least in part explain why GEP-NENs only in part response to anti-VEGF(R) therapy. Nonetheless, these findings also suggest that there is a specific population of GEP-NEN patients who will respond to a multikinase inhibitor therapy. Identification of benefitting subgroups might be feasible by immunohistochemistry as demonstrated.

Moreover, the identification of subgroups by using biomarkers is an evolving field in oncology and personalized therapy its goal [8,45]. Thus, staining for VEGFR seems to be a promising approach to detect patients who might profit from a multikinase inhibitor like pazopanib [33], cabozantinib [46,47], lenvatinib [46,47] or a highly selective VEGFR inhibitor such as axitinib [48]. However, up until now no specific VEGFR seems to be appropriate selecting patients for anti-angiogenic treatment. Axitinib is an orally bioavailable specific inhibitor of VEGFR 1-3 and currently approved for the therapy of advanced renal cell carcinoma. In a phase II study Strosberg et al. demonstrated antitumor activity of axitinib and a prolonged progression free survival of at least 14.6 months in patients with advanced GEP-NENs [48]. However, this novel tyrosine kinase inhibitor was associated with a high rate of toxicity. Therefore, the results of the present study might help to stratify patients who most probably will benefit from a therapy with axitinib in a future study.

Recently it has been demonstrated in renal cell cancer that the combination of axitinib and the PD-1 inhibitor embrolizumab results in significantly longer survival rates compared to monotherapy with sunitinib [49]. Treatment of GEP-NEN patients with immune checkpoint markers is currently under investigation and promising early results have been published [23,50]. Therefore, a further subject of the present analysis was to evaluate a potential coexpression of VEGFRs and immune checkpoint markers. It could be demonstrated that a high expression of PD-L1 is associated with VEGFR positivity in 9.8–25%. In this respect, further prospective studies are needed to evaluate a combination therapy approach in GEP-NEN patients by for example combining anti-VEGF compounds with anti-PD(L)-1 checkpoint inhibitors. Our data provide evidence that there may be a significant cross-talk between the VEGF/R and PD-1 pathway in GEP-NENs; this observation will add a scientific rationale for investigating combination approaches, as recently shown in hepatocellular carcinoma for the efficacy of the novel combination of bevacizumab and atezolizumab [17]. Currently a phase II study (NCT03290079) is recruiting patients with advanced GEP-NENs analyzing the efficacy of the combination of the anti-angiogenic agent lenvatinib with the checkpoint inhibitor pembrolizumab. This combination therapy has already proofed to be beneficial in various solid tumors, such as renal cell carcinoma, endometrial cancer, and urothelial cancer [16]. Moreover, the combination of Fosbretabulin, a compound in a class of agents termed vascular disrupting agents, with the approved Everolimus seems to be beneficial in the treatment of metastatic GEP-NEN patients in a phase 1 study [51].

Although this is a retrospective analysis, the largest cohort of patients with GEP-NENs regarding the expression of VEGFR 1-3 could be analyzed. Moreover, more than 90% of the analyzed patients underwent surgery representing a selection bias. However, anti-angiogenic treatment is considered primarily for patients with advanced disease and these patients are most probably no candidates for surgical resection. Thus, the present findings have to be interpreted with caution. Due to the sample size and the comprehensive analysis, this study gives a proficient overview of GEP-NENs. Furthermore, since patients were included during 18 years, the understanding and therapy of GEP-NENs have changed. Moreover, not every patient included has been treated at the Ludwig-Maximilans-University Munich. Thus, data on medical treatment could not be considered for survival analyses.

In conclusion, within this study a large well-characterized patient collective was classified for the first time according to the expression levels of VEGFR 1-3 and correlated to clinicopathological parameters. However, there was no prognostic benefit of a VEGFR based classification in GEP-NENs. Nonetheless, the present study provides evidence that a combination therapy of immune checkpoint markers and VEGFR inhibitors might be a promising approach. However, further prospective studies are necessary to evaluate the possible stratification of GEP-NENs regarding VEGFRs and the clinical value.

## Figures and Tables

**Figure 1 jcm-09-03368-f001:**
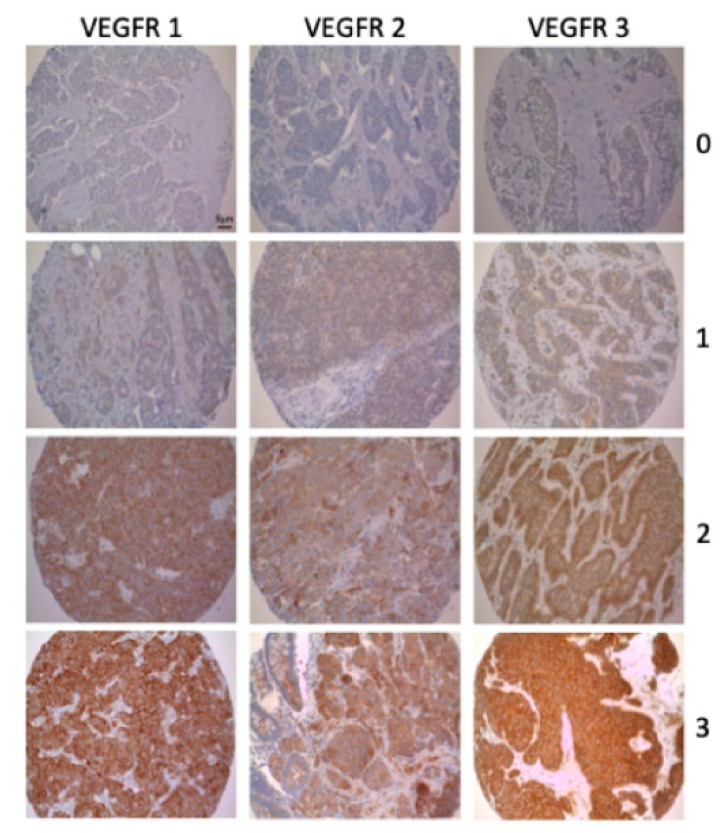
Examples of the immunohistochemical evaluation of VEGFR 1-3 (10× magnification): 0 = negative; 1 = weakly positive; 2 = moderately positive; 3 = strongly positive.

**Figure 2 jcm-09-03368-f002:**
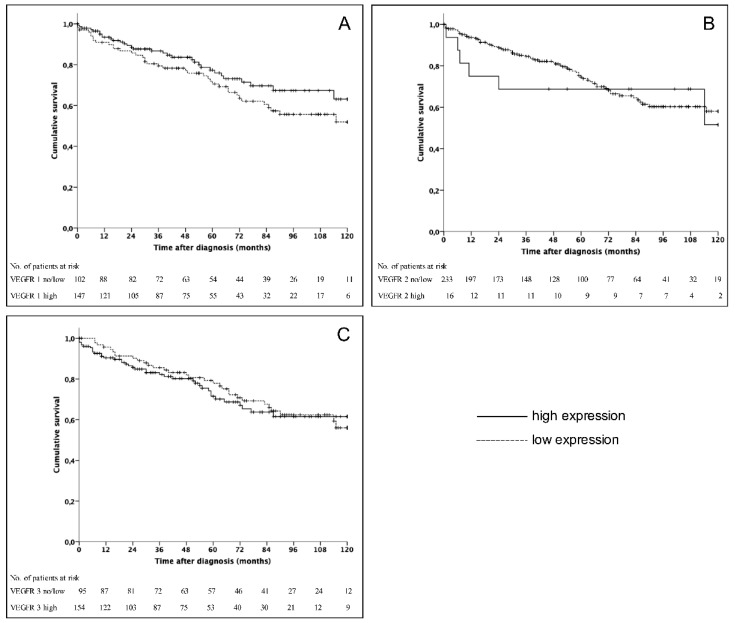
Kaplan–Meier survival curves for patients divided into VEGFR 1 (**A**) (*p* = 0.15), VEGFR 2 (**B**) (*p* = 0.723), and VEGFR 3 (**C**) (*p* = 0.708).

**Table 1 jcm-09-03368-t001:** Baseline characteristics in 249 patients with gastroenteropancreatic neuroendocrine neoplasms (VEGF: Vascular endothelial growth factor receptor; ^(1)^ 10 missing, ^(2)^ 23 missing, ^(3)^ 5 missing).

Patient Characteristics	*n*	%
Gender		
Male	137	55
Female	112	45
Localization		
Esophagus/stomach	16	6.4
Small intestine	126	50.6
Colon	18	7.2
Rectum	8	3.2
Papilla vateri	1	0.4
Pancreas	64	25.7
Appendix	16	6.4
VEGFR 1		
No/low expression	102	41
High expression	147	59
VEGFR 2		
No/low expression	233	93.6
High expression	16	6.4
VEGFR 3		
No/low expression	95	38.2
High expression	154	61.8
PD-L1 ^(1)^		
Low expression	218	91.2
High expression	21	8.8
PD-1 ^(2)^		
Low expression	188	83.2
High expression	38	16.8
Grading		
G1	152	61
G2	81	32.5
G3	16	6.4
Surgical resection ^(3)^		
Yes	229	92
No	20	8

**Table 2 jcm-09-03368-t002:** Different expression rates of VEGFR 1-3 regarding the primary tumor localization and the correlation of the expression rates of VEGFR 1-3 (n.a.—not applicable).

Localization (*n*)	VEGFR 1 High *n* (%)	VEGFR 2 High *n* (%)	VEGFR 3 High *n* (%)
Esophagus/stomach (16)	6 (37.5)	4 (20)	11 (68.8)
Small intestine (126)	74 (58.7)	3 (2.4)	64 (50.8)
Colon (18)	5 (28.7)	3 (16.7)	11 (61.1)
Rectum (8)	6 (75)	1 (12.5)	7 (87.5)
Papilla vateri (1)	0 (0)	0 (0)	1 (100)
Pancreas (64)	52 (81.3)	5 (7.8)	52 (81.3)
Appendix (16)	4 (25)	0 (0)	8 (50)
VEGFR 1 high	n.a.	9 (6.1)	121 (82.3)
VEGFR 2 high	9 (6.1)	n.a.	15 (93.8)

**Table 3 jcm-09-03368-t003:** Correlation of VEGFR 1-3 and PD-1 (programmed death-1).

	PD-1 Low Expression *n* (%)	PD-1 High Expression *n* (%)	*p*-Value
VEGFR 1	74	17	0.588
no/low expression	81.3%	18.7%	
VEGFR 1	114	21	
high expression	84.4%	15.6%	
VEGFR 2	183	32	0.004
no/low expression	85.1%	14.9%	
VEGFR 2	5	6	
high expression	45.5%	54.5%	
VEGFR 3	79	13	0.47
no/low expression	85.9%	14.1%	
VEGFR 3	109	25	
high expression	81.3%	18.7%	

**Table 4 jcm-09-03368-t004:** Correlation of VEGFR 1-3 and PD-L1 (programmed death ligand-1).

	PD-L1 Low Expression *n* (%)	PD-L1 High Expression *n* (%)	*p*-Value
VEGFR 1	89	7	0.643
no/low expression	92.7%	7.3%	
VEGFR 1	129	14	
high expression	90.2%	9.8%	
VEGFR 2	209	18	0.077
no/low expression	92.1%	7.9%	
VEGFR 2	9	3	
high expression	75.0%	25.0%	
VEGFR 3	91	1	0.001
no/low expression	98.9%	1.1%	
VEGFR 3	127	20	
high expression	86.4%	13.6%	

**Table 5 jcm-09-03368-t005:** Cox proportional hazard regression analysis.

	*p*-Value	Exp (B)	CI 95%
Distant metastases	0.042	2.09	1.029–4.246
Grading	0.000	1.036	1.024–1.048
Bone metastases	0.006	3.242	1.398–7.516
Liver metastases	0.542	1.185	0.687–2.043

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
