# Peer review of "Distinct Expression Patterns of VEGFR 1-3 in Gastroenteropancreatic Neuroendocrine Neoplasms: Supporting Clinical Relevance, but not a Prognostic Factor"

_jcm, 2020, doi:10.3390/jcm9103368_

Round 1

Reviewer 1 Report

The authors have adequately and thoroughly responded to my comments and suggestions. Please see minor suggestions: 

Abstract introduction

Update the aim so that it corresponds with what is stated in the Introduction of the manuscript. 

Results

Table 2: You could omit the last row (VEGF3 high) as those results are provided in the rows just above.

Survival analyses: Please provide hazard ratios and 95% confidence interval to allow interpretation of the Cox regression analyses. I would guess that e.g. age has a hazard ratio >1, whereas resection of primary has a hazard ratio < 1. Without the hazard ratios, it is impossible to interpret the results.

Discussion

Line: 268-270: “Only few G3 GEP-NENs were included in the present analysis and these tumors showed only rare VEGFR 1- 3 expression”. These data are  not available under “Results”.

Reviewer 2 Report

All the raised issues have been addressed, I have no further comment

Author Response

This manuscript is a resubmission of an earlier submission. The following is a list of the peer review reports and author responses from that submission.

Round 1

Reviewer 1 Report

The manuscript by Bosch et al investigates the role of VEGFR expression in a large series of neuroendocrine neoplasia, focusing on the potential role of these markers to modify clinical outcome.

The study has the merit to cover an interesting and debated (although not so original)  topic, however it presents several major limitations which need to be carefully considered.

  1. the aim of the study should be better defined, since it sounds generic as is. Authors should clearly state what was the purpose of their investigation.
  2. The study design (I suppose retrosepctive) should be declared, as well it should be specified which WHO classification was used to re-classify tumors (I suppose WHO 2017/2019, citation needed)
  3. Statistical methods. Specify which test was used to perform  analysis to asess predictors for poor survival
  4. Specify how many tumors were NET G3 vs NEC G3 in the group of G3 NENs, according with the recent WHO classifications. Give the median value (and range) of Ki67 for the entire cohort of tumors.
  5. Specify the site of distant metastases, with particular effort on bone metastases which are well known to be significant prognostic factors in advanced NENs (Scharf et al Neuroendocrinology 2018, Panzuto et al Pancreatology 2019)
  6. Give data on medical treatments which patients had received (if any) before VEGF evaluation and during follow-up. Thi is crucial to understand data on survival
  7. Try to perform multivariate analysis when searching for predictors (para on survival analysis). Try to test Ki67 as continuous variable in addition to G cathegories. Again, analyse specific metastatic site (extra-hepatic, bone...) to understand their impact on survival
  8. Add the "number at risk" under the survival curves
  9. I don't agree with the study conclusions. Authors state that "The expression levels of the three VEGFRs do not represent a prognostic factor" (line 233) and that "There was no significant association of the expression levels of VEGFR 1-3 and overall survival" line 225. I believe these are the major findings, which should be reported at the beginning of the discussion, as well as in the conclusions. I also should consider to change the study title by focusing on this "negative" result, which however might be helpful for physicians dealing with NENs
  10. There is no reported data supporting the suggestion to promote combined therpay with VEGFR inhibitors and immunotherapy. The conclusion reported in lines 264-265 is not supported by evidence, and should be deleted, or at least moved in the discussion as a hypothesis which need to be verified by specific study.

Author Response

Please see the attached response letter.

Reviewer 2 Report

The authors are to be congratulated on the work of presenting expression patterns of VEGFR and PD(L)-1 in an extensive series of GEP-NEN patients.  In the current form however a number of issues should be considered:

Introduction:

  • It is stated that the overall the aim is to describe the VEGFR expression in GEP-NENs and the prognostic role of this. However, first in the Methods PD(L)-1 is also described. The rationale and aim of including PD(L)-1 in this work should be clearly stated in the Introduction.
  • Line 72-73: ..”to describe the different emphasis of these receptors in GEP-NENs” needs to be clarified. Please clarify - how should this be understood?

Methods:

  • Patients were included between 1995-2013. Treatment of this patient group has changed considerably. Did any of the patients receive anti-angiogenic treatment during follow-up? This may be an interesting subgroup for analyzing response to treatment depending on expression pattern of VEGFR
  • Line 77: Most patients underwent surgery according to Table 1. Was the ”..GEP-NEN specimens..” obtained from that procedure or biopsies?. As time from diagnose is used in the survival analyses it would be relevant to report the time elapsed from diagnosis to the GEP-NEN specimen was obtained.
  • Line 82: Please include reference to WHO classification used.
  • I would recommend to report the results of the VEGFR grading, so that the actual numbers of negative, weakly positive, moderately positive and strongly positive cases are reported separately. This could be in a Supplementary file. Also it does not seem appropriate to use the term “low expression”, when this refers to both negative and weakly positive staining.
  • The grading of PD(L)-1 is not described and in Table 1 is referred to as “no expression vs expression”, but in Table 3 and 4 as “low vs high expression”

Results:

In general the P-values should be reported exactly, e.g. P =0.13 (not P > 0.05 and P<0.05). P<0.001 is generally accepted.

  • Line 124: A statistical test of women vs. men is not meaningful.
  • Line 131: 53% of 186 patient is reported as 132 cases? How is that to be understood?
  • Table 2: Data for VEGFR 1 vs. VEGFR 2 is not reported. However, the rationale for investigating association between the receptors is unclear to me. It would be more useful to see the data for different tumor locations presented as a table.
  • Line 158: I believe a typographical error is present. N=15 is written, but I guess it should be n=51?
  • Figure 3: Only data regarding specific subtypes are presented. Either data for the other subgroups e.g. G2 tumors and G3 tumors should be presented or not at all. The finding of a statistical trend in such a subgroup analysis should at least be interpreted cautiously.

Discussion:

  • Please add a discussion as to why VEGFR 2 was only expressed in 6% vs. around 55% for receptor type 1 and 3.
  • Consider mentioning other biological markers for selection of patients for anti-angiogenic treatment e.g. obtained from imaging markers (e.g.: Chaan S. Ng et al: CT perfusion in normal liver and liver metastases from neuroendocrine tumors treated with targeted antivascular agents).
  • Considering the large variation in the expression of VEGFR 1-3, which would you argue should be used for selection of patients for anti-angiogenic treatment?
  • Line 210: “…unpleasant response rate”. Please clarify.
  • Line 235: “…correlation of the expression levels of VEGFR 1-3 with tumor progression”. This is presented in the results. Should this be rephrased as tumor stage?.

Limitations:

  • More than 90% of the included patients underwent surgery. As such a selection bias must be expected. Does this reflect the patient population that is considered for anti-angiogenic treatment?

Author Response

(The authors gave the same response as above.)
